# Cross-organ analysis reveals associations between vascular properties of the retina, the carotid and aortic arteries, and the brain

Sofía Ortín Vela [1,2] ✉ & Sven Bergmann [1,2,3] ✉

## Abstract

**Background** Vascular properties of the retina are indicative of both ocular and systemic cardio- and cerebrovascular health. However, the specific relationships between retinal and non-retinal vascular phenotypes have not been systematically investigated in large samples. This study aims to compare cross-organ phenotypic and genetic relationships between vascular characteristics across different body sites.

**Methods** We compared vascular image-derived phenotypes from the brain, carotid artery, aorta, and retina, using UK Biobank sample sizes ranging from 18,808 to 68,740 participants. We examined phenotypic and genetic correlations, as well as common associated genes and pathways.

**Results** Here we show that white matter hyperintensities are positively correlated with carotid intima-media thickness ($r = 0.03$), lumen diameter ($r = 0.14$), and aortic cross-sectional areas ($r = 0.09$), but negatively correlated with aortic distensibilities ($r \leq -0.05$). Arterial retinal vascular density shows negative correlations with white matter hyperintensities ($r = -0.04$), intima-media thickness ($r = -0.04$), lumen diameter ($r = -0.06$), and aortic areas ($r = -0.05$), while positively correlating with aortic distensibilities ($r = 0.04$). Significant correlations also persist after correcting for hypertension.

**Conclusions** Our findings shed light on the complex interplay between vascular morphology across different organs, revealing both shared and distinct genetic underpinnings. Retinal vascular features reflect broader systemic vascular morphology and offer an accessible window into cardio- and cerebrovascular health.

## Plain Language Summary

Doctors often use eye scans to check for signs of heart and brain disease, but the exact link between the tiny blood vessels in the eye and those in major organs is unclear. We aimed to systematically map similarities between blood vessels across the entire body. We analyzed medical scans of the retina, brain and major arteries (carotid and aorta) from a large group of people from the UK. We looked for similarities between blood vessel properties in different organs and studied to what extent they are inherited. We found strong connections with the health of retinal blood vessels mirroring the health of the brain and major arteries. This suggests that some of the same factors influence vessel health across the body. This suggests that an eye scan could be a fast, non-invasive way to get a complete snapshot of a person's overall cardiovascular and brain health. These findings could help doctors identify health issues, such as early artery stiffness or brain aging, much sooner.

The vascular system ensures the distribution of oxygen and nutrients and the removal of waste through a hierarchical network of arteries, capillaries, and veins. While basic physiological measures, such as blood pressure, provide global insights into vascular function, medical imaging enables the quantification of detailed, organ- and vessel-specific phenotypes.

Invasive techniques like contrast-enhanced angiography provide high-resolution images of blood vessels, but they are typically used only when a disease risk is suspected[1]. Non-invasive techniques like non-contrast

magnetic resonance angiography (MRA) are better suited for population studies but incur high costs and have lower spatial resolution[2]. In contrast, retinal imaging is inexpensive, non-invasive, and routinely available, making it a practical tool for examining systemic vascular health[3].

Changes in vascular image-derived phenotypes (IDPs) have been linked to cardiovascular risk across multiple organs. For example, increased arterial stiffness and carotid intima-media thickness (IMT) are known risk factors for hypertension and cardiovascular diseases[4]. Higher carotid IMT is

[1]Department of Computational Biology, University of Lausanne, Lausanne, Switzerland. [2]Swiss Institute of Bioinformatics, Lausanne, Switzerland. [3]Department of Integrative Biomedical Sciences, University of Cape Town, Cape Town, South Africa. ✉e-mail: sofia.ortinvela@unil.ch; sven.bergmann@unil.ch

also related to atherosclerosis[4], while a larger carotid lumen diameter (LD) is associated with a higher risk of cardiovascular events[5,6]. Moreover, abdominal aortic, intracranial, and coronary artery aneurysms share similar pathophysiological mechanisms involving vascular dilation and remodeling[7], suggesting a common underlying vascular wall pathology[7]. In addition, aortic sizes and distensibilities are also risk factors of aneurysms and other cardio- and cerebrovascular diseases[8]. Similarly, changes in the retinal vasculature have been linked to vascular issues in distant organs, including stroke[9–11], coronary heart disease[12,13], or hypertension[14,15], highlighting systemic links central to oculomics, an emerging field that uses retinal imaging to assess overall systemic health[16].

Despite this evidence, the relationships between similar vascular properties across different organs are still poorly understood. Previous studies have analyzed phenotypic and genetic correlations between multi-organ IDPs, including those of the heart, brain, retina, and liver[17–20], but they have not specifically addressed vascular IDPs, which capture vessel morphology and function.

This study aims to shed light on these associations using data from the UK Biobank (UKB)[21,22], which includes multiple vascular IDPs from the brain, carotid[23], aorta[24–29], and retina[30,31]. We investigated both phenotypic and genetic correlations between these IDPs, highlighting shared associated genes and pathways. Our approach reveals several cross-organ relationships between vessel morphology and function phenotypes, including those of the retina, emphasizing its potential as a prognostic proxy for non-retinal vascular diseases.

## Methods

### The UKB imaging study

The UKB is a large-scale biomedical database and research resource containing anonymised genetic, lifestyle, and health information from its participants. The UKB database includes blood samples, heart and brain scans, and genetic data of the volunteer participants, is globally accessible to approved researchers who are undertaking health-related research that is in the public interest. UKB recruited 500k people aged between 40 and 69 years in 2006–2010 from across the UK. With their consent, they provided detailed information about their lifestyle, physical measures and had blood, urine, and saliva samples collected and stored for future analysis. It includes multi-organ imaging for many participants, such as MRI scans of the brain, heart, and liver, carotid ultrasounds, and retinal CFPs[32].

The brain imaging data, covers six modalities: T1-weighted structural MRI, T2 FLAIR (fluid-attenuated inversion recovery), susceptibility-weighted MRI, resting-state functional MRI, task functional MRI, and diffusion MRI. These modalities yield various[33], including: a) Mean cerebral blood flow (CBF) maps, derived from arterial spin labeling (ASL) perfusion MRI, they provide insights into resting cerebral blood flow across the brain. b) Mean arterial transit time (ATT) maps, also from ASL data, and they indicate the time it takes for blood to travel from the neck to a given region of interest, potentially revealing vascular issues. Measures of mean CBF and ATT are available for each brain region. c) WMH volumes, obtained from T2 FLAIR structural MRI scans, they serve as markers for cerebral small vessel disease. The mean intensity and volume of vessels are obtained from T1-weighted brain MRI using the Freesurfer automatic subcortical segmentation (ASEG) tool, for the right and left hemispheres ('Brain Imaging Documentation').

Carotid ultrasound data, available for ~50k participants, were collected to measure carotid IMT, a marker for subclinical atherosclerosis and cardiovascular disease risk. Images were acquired from both left and right carotid arteries using standardized protocols across all assessment centers. Measurements were taken at four angles (120°, 150°, 210°, and 240°) around the carotid bulb. For each angle, the maximum, mean, and minimum IMT values were computed ('Carotid Ultrasound Documentation'). It is important to note that IMT values were available for more participants than the carotid images themselves (only ~20k).

Cardiovascular MRIs were acquired on 1.5T Siemens MAGNETOM Aera scanners. The imaging protocol included several sequences: Bright blood anatomical imaging in sagittal, coronal, and transverse planes; cine imaging of the left and right ventricles in both long-axis and short-axis views; myocardial tagging for strain analysis; native T1 mapping; aortic flow quantification; and imaging of the thoracic aorta[34]. This comprehensive protocol allows for detailed evaluation of aortic geometry and function, providing (among others) IDPs related to aortic distensibility and dimensions. Aortic distensibility, which reflects aortic stiffness, was measured directly by the relative change in aortic cross-sectional area per unit change in pressure (lower distensibility signifies increased stiffness). Additionally, a variety of aortic dimension measures, including maximum and minimum areas, mean and standard deviation areas during diastole and systole, and mean absolute deviation, capture the dynamic changes in the cross-sectional areas of the ascending and descending aorta throughout the cardiac cycle[24–27].

Retinal CFPs, available for around 90k participants, were acquired from both eyes using a Topcon 3D OCT 1000 Mark II camera, with images centered to include both the optic disc and macula within a 45° field-of-view.

Access to the UKB data was granted under application number 90947. The UKB study was approved by the North West Multi-centre Research Ethics Committee (MREC). No additional institutional review board (IRB) approval was required for this study because the use of UKB data is covered by the original, generic approval granted by the MREC, and this access is for secondary analysis of de-identified, publicly-available data, as stipulated on the UKB website.

### Main IDPs selection

We identified imaging modalities capable of capturing morphological or functional vascular IDPs. Our search combined a systematic review of the UKB database, Google Scholar, and the GWAS catalog, using both general vasculature-related keywords (e.g., 'vessel', 'vascular', 'vasculature', 'blood', 'artery', 'arterial', 'arterioral', 'vein', 'venular') and specific vessel names (e.g., 'carotid', 'aorta'). Although our search aimed to capture all morphological and functional vascular IDPs within the UKB dataset, we also found additional vascular IDPs in external sources that, although measured on the UKB dataset, were not yet available (see Figshare[35] for more details).

In the initial selection, we selected all organ-specific geometric and functional images. For the brain, several IDPs were available, including CBF and ATT for all parts of the brain. To manage the high correlation among these measures, we calculated the CBF and ATT averaged across the brain. We also considered total/deep/and peri-ventricular WMH volumes, as well as the mean intensity and volume of vessels in the brain, for both hemispheres (see Supplementary Figs. 1, 2). Carotid IMT IDPs were measured at four different angles, and for each, the main, minimum, and maximum values were available. These values were averaged across angles for consistency with previous studies. For heart vascular IDPs, we included various measures of ascending and descending areas, systolic and diastolic parameters, standard deviation, mean absolute deviation, stroke, and ejection fraction (see Supplementary Fig. 2).

Further simplification was applied while ensuring comprehensive representation of vascular morphology and functionality across all organ imaging modalities. For the brain, although CBF and ATT are informative for functional associations, they were excluded as main IDPs due to their limited sample size. Among the three WMH IDPs, only total WMH was retained, given its strong correlations with other IDPs and its well-established role as a marker of small vessel disease[36]. Although FreeSurfer (automatic segmentation of T1 images)-derived IDPs of 'vessel' volume and intensity were available, we did not selected them since the number of voxels typically attributed to these regions was extremely small, and the corresponding IDPs did not correlate significantly with any of the other IDPs we studied (see Supplementary Fig. 2). This suggests they may, at best, be very noisy representations of vascular entities.

For the carotid artery, only the minimum IMT was chosen due to its strong correlation with other measures. For the aorta, we selected the minimum areas and distensibilities of the ascending and descending aorta because of their relevance and high correlations with other IDPs.

**Table 1 | Demographic details and sample sizes for main non-retinal IDPs**

| IDP | Sample size | Mean age (y) | Std age (y) | Female/Male |
|---|---|---|---|---|
| WMH total volume | 40,302 | 64.0 | 7.7 | 1.11 |
| Min carotid IMT | 49,282 | 64.5 | 7.8 | 1.07 |
| Carotid LD | 18,808 | 64.0 | 7.6 | 1.03 |
| Asc aorta distensibility | 32,962 | 63.6 | 7.5 | 1.05 |
| Asc aorta min area | 36,120 | 63.6 | 7.6 | 1.06 |
| Desc aorta distensibility | 32,970 | 63.6 | 7.5 | 1.05 |
| Desc aorta min area | 36,121 | 63.6 | 7.6 | 1.06 |

Sample size (number of participants), mean age (in years), standard deviation of age (in years), and the female-to-male ratio for the main non-retinal IDPs.

Finally, for retinal vascular IDPs, we aimed to ensure consistency with other morphological vascular IDPs, selecting only vascular density, tortuosity, and vessel diameters to comprehensively represent vascular morphology across different organs and systems.

## Phenotypic correlation

For retinal IDPs, we used data from instance 0, which corresponds to the initial assessment visit (2006–2010) when participants were recruited and consent given, and instance 1, which corresponds to the first repeat assessment visit (2012–13). Whenever retinal images from both eyes were available, the retinal IDP was computed for each eye, and the average was taken; otherwise, the IDP from the available image (right or left) was used. These IDPs were z-scored and adjusted for various covariates, including sex, age, age-squared, sex-by-age, sex-by-age-squared, spherical power, spherical power-squared, cylindrical power, cylindrical power-squared, instance, assessment center, genotype measurement batch, and genetic PCs 1–20[31].

For the non-retinal vascular IDPs, only instances 2 (imaging visit, 2014+) and 3 (first repeat imaging visit, 2019+) were available. We used instance 2 due to its larger sample size, as fewer than 300 individuals per IDP were added if we included instance 3. Retinal IDP outliers were removed using a 10 std threshold[31]. The same criterion was applied to the non-retinal IDPs. These IDPs were also z-scored and adjusted for covariates, including age attended ('21003-2'), age attended squared, sex ('31'), UKB assessment center ('54'), standing height ('50-2'), and genetic PCs 1-20 ('22009-0.1' to '22009-0.20'). LD measures were obtained as described in[23] and were processed similarly to the other IDPs. The inclusion of genetic PCs in both retinal and non-retinal IDP adjustments aimed to account for population structure without restricting the analysis to individuals of European ancestry. This allowed us to retain a larger and more diverse sample while mitigating the confounding effects of ancestry.

We reduced the selected non-retinal IDPs as detailed in Section 2.2. After obtaining the residuals for all IDPs, we performed phenotypic correlation analysis using Pearson's correlation coefficient. Multiple testing correction was applied using the following significance levels: *: $p < 0.05/N_{test}$, **: $p < 0.001/N_{test}$, where $N_{test} = N_{IDPs} \times (N_{IDPs}/2 + N_{retina})$.

For the main non-retinal IDPs, the sample sizes and demographics can be found in Table 1:

## Genetic correlation

To analyze genetic correlations for the main vascular IDPs, we accessed published GWAS summary statistics from the UKB. For the brain, we used summary statistics for the total volume of WMH, adjusted for covariates such as age, age-squared, sex, sex-by-age, sex-by-age-squared, 10 genetic PCs, head size, head position in the scanner, scanner table position, assessment center location, and date of attending assessment center. The sample size for these statistics was ~33k participants[37], 'Brain summary statistics'.

For the carotid minimum IMT, summary statistics were adjusted for the covariates: age at the time of the imaging visit, sex, genotyping array, and 30 genetic PCs, with a sample size of around 44k participants[38]. Summary statistics for the ascending and descending aorta distensibility and minimum areas were available in ref. 24, 'Aorta summary statistics', using *bolt_P_BOLT_LMM_INF*. Retina summary statistics used the same covariates as those applied in the phenotypic analysis[31], 'Retina summary statistics'. And similarly for the carotid LD[23]. More information about the main IDPs can be found in Figshare[35].

Genetic correlations and $h^2$ were computed using LDSR, using the 1000G EUR reference panel. Detailed results for $h^2$ can be found in Figshare[35]. LDSR was also used to derive the genetic correlations between IDPs[39,40].

## Genes and pathways

Gene and pathway scores were computed using *PascalX*[41,42]. Both protein-coding genes and lincRNAs were scored using the approximate "saddle" method, taking into account all SNPs within a 50kb window around each gene. All pathways available in MSigDB v7.2 were scored using *PascalX* ranking mode, fusing and rescoring any co-occurring genes less than 100kb apart. *PascalX* requires LD structure to accurately compute gene scores, which in our analyses was provided with the UK10K (hg19) reference panel. Correction for bias due to sample overlap was done using the intercept from pairwise LDSR genetic correlation. The significance threshold was set at 0.05 divided by the number of tested genes (RANKING).

Gene-level cross-GWAS coherence test between IDP pairs and between IDPs and diseases or risk factors was computed using the *PascalX* cross-scoring zsum method[41], testing for both coherence and anti-coherence of GWAS signals.

## Results

The UKB includes nearly 186k retinal color fundus photographs (CFPs) from ~90k participants. We recently derived 17 different IDPs for 71k participants whose CFPs passed quality control (QC)[31], which we used in this investigation. In addition, we developed an automated method to measure the carotid LD from ultrasound still images, allowing us to assess it for 18,808 UKB participants, after QC[23]. Furthermore, we identified other reliable non-retinal vascular IDPs available for a substantial number of participants (≥1k), specifically: (1) white matter hyperintensities (WMH) from T2-weighted brain magnetic resonance imaging (MRI), (2) measurements of carotid IMT from ultrasound images, and assessments of the (3) cross-sectional area and (4) distensibility of the ascending and descending aorta from cardiac MRI (see Fig. 1). After filtering, these IDPs were available for ~40k, 49k, 33k, and 36k participants, respectively (c.f. Figshare[35]).

For retinal IDPs, we focused on tortuosity, vascular density, and vessel diameter, which can be compared to the non-retinal IDPs we used. While retinal IDPs include both arteriolar and venular measurements, non-retinal IDPs focus mainly on arterial characteristics (see Figshare[35] for details).

### Phenotypic and genetic correlations between vascular IDPs

We first adjusted all vascular IDPs by regressing out common covariates, including sex, age, age-squared, assessment center, standing height, and the first 20 genetic principal components (PCs). We then calculated pairwise correlations between the corrected IDPs. The left panel of Fig. 2a shows the phenotypic correlations among IDPs of the brain, carotid, and aorta. The total volume of WMH was positively correlated with the carotid IMT ($r = 0.03$, $p \leq 10^{-10}$), the LD ($r = 0.14$, $p \leq 10^{-66}$), and the aortic areas ($r = 0.09$, $p \leq 10^{-60}$), but negatively correlated with the aortic distensibilities ($r \leq -0.05$, $p \leq 10^{-17}$). IDPs related to aortic areas and distensibilities showed strong positive correlations between the ascending and descending aorta, respectively ($r = 0.44$, and $r = 0.74$, $p \leq 10^{-200}$), but were negatively correlated with each other ($r \leq -0.14$, $p \leq 10^{-143}$). These measures were at best weakly correlated with the carotid IMT. However, a similar pattern was

**Fig. 1 | Visualization of the human vascular system and some of the imaging modalities available in the UKB.** The modalities include CFPs for examining the retinal vasculature, ultrasound for assessing the common carotid arteries (in particular, the IMT and the LD), transverse cardiac MRI for evaluating the aorta, and structural brain MRI for detecting WMH.

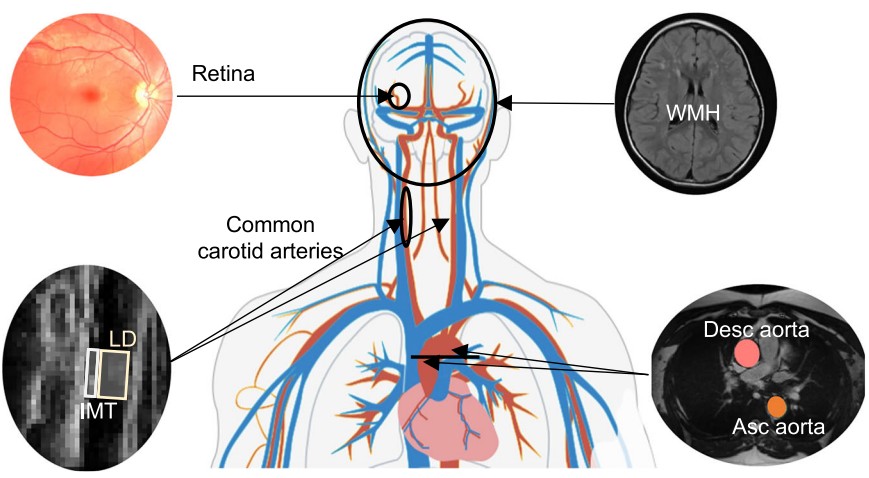

observed when correlating with carotid LD, with stronger signals compared to those seen with IMT, despite having a much smaller sample size (see Supplementary Fig. 3).

The right panel of Fig. 2a displays correlations between these IDPs and retinal vascular IDPs. The strongest positive correlations were found between the distensibility of both the ascending and descending aorta and the arteriolar retinal vessel diameter measurements (both the median across all segments and the central retinal equivalent, which is specific to the largest retinal blood vessels; $r \geq 0.05, p \leq 2 \times 10^{-6}$). In contrast, the aortic areas were negatively correlated with these retinal IDPs ($r \leq -0.04, p \leq 3 \times 10^{-5}$) and with the retinal arteriolar vascular density ($r = -0.05, p \leq 1 \times 10^{-5}$). Notably, retinal arteriolar vascular density was the only retinal IDP significantly correlated with the carotid IMT ($r = -0.04, p \leq 3 \times 10^{-5}$) and LD ($r = -0.06, p \leq 3 \times 10^{-6}$). We also observed significant correlations between WMH and several retinal IDPs, with the strongest negative correlation seen with arteriolar median diameter ($r = -0.05, p \leq 2 \times 10^{-6}$) and, somewhat weaker but still significant, with the arteriolar central retinal equivalent ($r = -0.04, p \leq 2 \times 10^{-5}$). Additionally, arteriolar tortuosity was negatively correlated with the area of the descending aorta ($r = -0.05, p \leq 3 \times 10^{-6}$). For more details, see Supplementary Note 1", and Supplementary Fig. 3, which shows the number of participants available for each IDP pair.

Because blood pressure strongly influences vascular properties globally[43], we also recomputed the pairwise correlations after adjusting for hypertension (see Fig. 2b). This additional adjustment reduced the magnitude of inter-IDP correlations overall, but did not affect the sign of any sizable correlation, apart from those involving the carotid IMT. Despite this reduction, the retinal arteriolar vessel diameter measures remained significantly positively correlated with the aortic distensibility and negatively with the area of the ascending aorta. Similarly, the negative correlation between the area of the descending aorta and arteriolar tortuosity persisted. However, the small but highly significant correlation between carotid IMT and aortic IDPs became non-significant. For further details, see Figshare[35].

To evaluate whether additional cardiovascular risk factors might further confound these relationships, we repeated the analyses including an extended covariate set with diabetes, current smoking, and HDL cholesterol, together with the previous covariates and hypertension status. Results remained consistent with the hypertension-adjusted analyses: effect sizes changed only modestly, and all of the previously observed strong correlations persisted (Supplementary Fig. 4).

Next, we computed cross-IDP genetic correlations and SNP-heritabilities ($h^2$) using Linkage Disequilibrium Score Regression (LDSR)[39,40], leveraging GWAS summary statistics from previous studies (see Section 2). We observed significant genetic correlations exclusively between IDPs of the same organ (see left of Fig. 2c, where $h^2$ are displayed on the diagonal). Specifically, the cross-sectional areas of the ascending and descending aorta were positively correlated with each other and negatively with the distensibilities. Furthermore, carotid LD was positively correlated

with IMT, as well as with the ascending and descending cross-sectional areas of the aorta. Correcting for multiple hypotheses testing, no significant genetic correlation was found between these IDPs and retinal vascular IDPs (see right panel of Fig. 2c). For more details on summary statistics, refer to the methods Section 2.4, and Figshare[35].

### Genes and pathways associated with vascular IDPs

To identify genes associated with these IDPs, we used the *PascalX* analysis tool[41,44]. The number of genes associated with non-retinal IDPs tended to be larger for IDPs with higher $h^2$ (see Fig. 3a, left, diagonal). The IDPs with more genes associated were the area of the ascending aorta ($h^2 = 0.36$; 157 genes) and the carotid IMT ($h^2 = 0.21$; 97 genes). The descending aorta ($h^2 = 0.28$; 90 genes) had fewer associated genes than its ascending counterpart. Distensibilities had much less associated genes, yet again more for the ascending ($h^2 = 0.09$; 12 genes) than the descending aorta ($h^2 = 0.08$; 7 genes). We also observed a large number of associated genes with WMH ($h^2 = 0.26$; 85 genes). The off-diagonal elements show the number of common genes for each pair of non-retinal IDPs. No single gene was shared among all non-retinal IDPs. WMH shared 22 genes with IMT, 18 with LD, and 11 with the ascending aorta area. IMT shared 32 genes with LD, 27 genes with the area of the ascending and 6 with the area of the descending aorta. Additionally, the respective genes associated with identical IDPs for the descending and ascending aorta displayed some overlap. The venular central retinal equivalent shared genes with WMH, IMT, LD, and the areas of the ascending and descending aorta (Fig. 3a, right), including genes like *FBN1*, *CAPN12*, *EIF3K*, and *SH2B3*. Retinal vascular densities shared genes with the descending aorta area, while tortuosities shared genes with WMH, IMT, LD, and the areas of the ascending and descending aorta, including genes like *SMAD3*, *COL4A1*, and *COL4A2*. For more details, see Figshare[35].

Further analyses using a local genetic correlation approach were conducted to identify shared genes among pairs of IDPs, moving beyond simple intersection of individually associated genes. For this, we employed *PascalX* cross-GWAS analysis[41], which allowed us to distinguish coherent (positively correlated) and anticoherent (negatively correlated) genetic effects. Notably, WMH and IMT exhibited mostly coherent genes, suggesting aligned genetic influences, whereas IMT and LD showed predominantly anticoherent genes, indicating opposing genetic effects. The venular central retinal equivalent showed exclusively coherent gene relationships with both WMH and IMT, while showing mainly anticoherent associations with LD and the descending aorta minimum area. Associations between tortuosities and the descending aorta minimum area were largely coherent. Detailed data on coherence and anticoherence relationships can be found in Figshare[35].

We also used *PascalX* to identify gene-sets (or "pathways") enriched with high-scoring genes[41,44]. The number of pathways associated with each IDP is shown in Fig. 3b, left, diagonal. Specifically, ascending (36) and descending aorta minimum area (18) had the most associated pathways, and were the only IDPs with shared pathways (4), including the '*GO HEART*

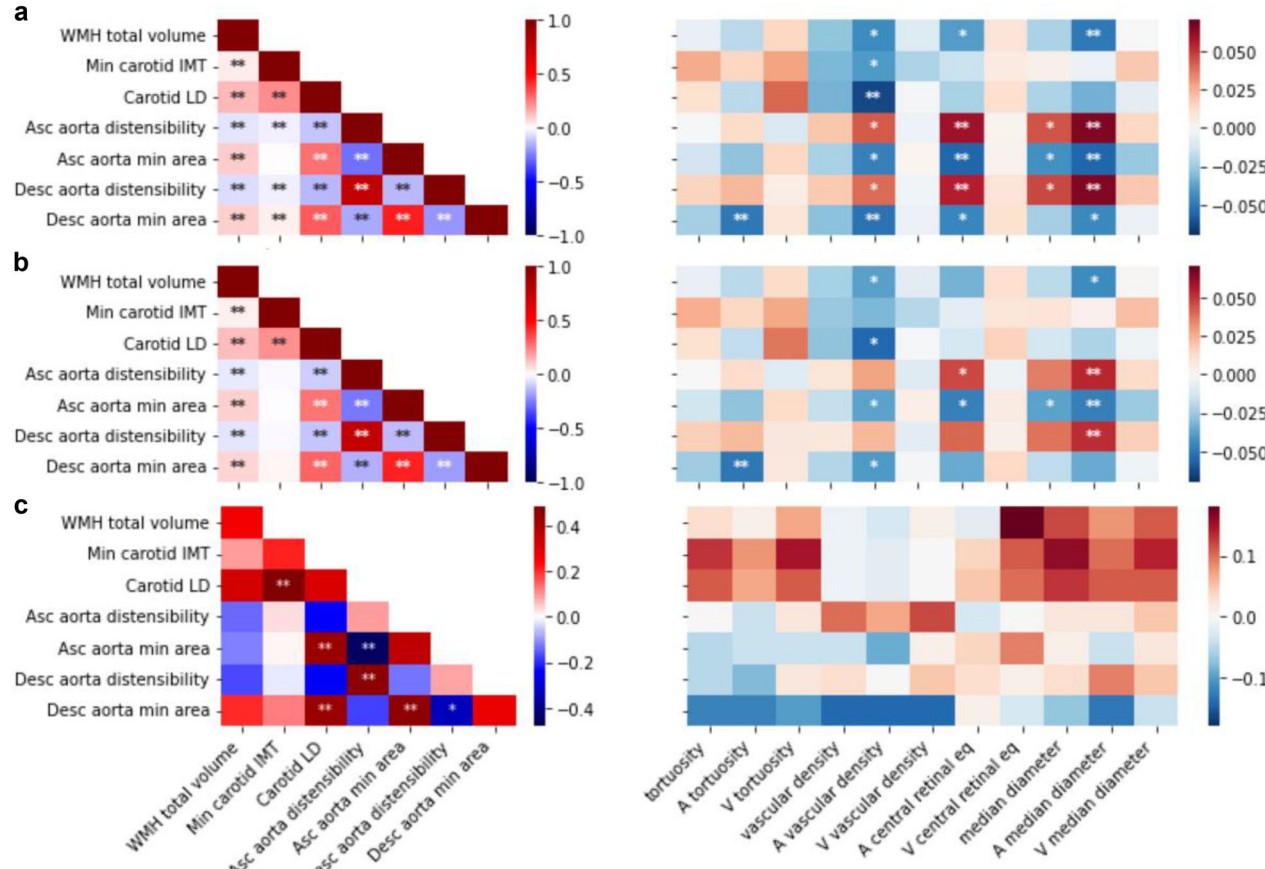

**Fig. 2 | Phenotypic and genetic correlation maps between vascular image-derived phenotypes across the brain, aorta, carotid artery, and retina. a** Phenotypic correlation of non-retinal IDPs with non-retinal IDPs (**left**) and retinal IDPs (**right**), all corrected for common covariates. **b** Phenotypic correlation of non-retinal IDPs with non-retinal IDPs (**left**) and retinal IDPs (**right**) additionally adjusted for hypertension. Hypertension was defined as having a systolic blood pressure (SBP) ≥140 mmHg or diastolic blood pressure (DBP) ≥90 mmHg. **c** Genetic correlation of non-retinal IDPs with non-retinal IDPs (**left**) and retinal IDPs (**right**), computed using LDSR[39]. The diagonal of the left figure shows the heritability values of the non-retinal IDPs (values can be found in Figshare[35]). GWAS summary statistics from previous studies were used to conduct this analysis (see Section 2). GWAS sample size for WMH was around ~33k participants, for IMT around ~44k, for aortic IDPs ~33–38k, and retinal IDPs ~54–69k. For all images, the x-axis shows non-retinal IDPs (**left**) and retinal IDPs (**right**) and the y-axis shows non-retinal IDPs. Colors indicate Pearson's correlation coefficients (**a, b**) or the genetic correlation coefficient (**c**). Significance for Pearson's correlations was tested using a two-sided test, and asterisks indicate the level of statistical significance (except for the diagonal of Figure **c**). These *p*-values were corrected for multiple testing (*$p < 0.05/N_{test}$, **$p < 0.001/N_{test}$, where $N_{test} = N_{IDPs} \times (N_{IDPs}/2 + N_{retina})$). All IDPs were adjusted for covariates (see section 2). For more details, see Figshare[35].

*DEVELOPMENT*' pathway. These IDPs also had the largest pathway overlap with retinal IDPs (Fig. 3b, right), mainly with those related to tortuosity. The venular central retinal equivalent shared one gene set with IMT and LD, specifically a gene cluster on '*chr8p23*'. Vascular density shared one pathway with the ascending aorta minimum area, while tortuosities shared pathways with WMH and the areas of the aorta, including pathways such as '*MANNO MIDBRAIN NEUROTYPES HENDO*', '*HP ABNORMAL VASCULAR MORPHOLOGY*', and actin-related ones, like '*GO ACTIN FILAMENT BUNDLE*', and '*GO ACTIN BINDING*'. For more details, see Figshare[35].

## Discussion

In this study, we explored cross-organ associations in vascular IDPs, including both morphological and functional measures. By leveraging the extensive imaging data from the UKB, we conducted an analysis that included IDPs from the brain, carotid, aortic arteries, and the retina. This comprehensive approach enabled us to discover numerous significant correlations both at the phenotypic and genetic level.

In our previous work with IDPs measured from retinal CFPs[31], we observed that morphological IDPs in the retina were often correlated. In this study, we found that also morphological measures of different arteries, specifically the cross-sectional areas of both the ascending and descending aorta, as well as the carotid LD, are all positively correlated with each other. We also observed positive correlations between WMH volume and carotid measures, further supporting previous reports that greater IMT and LD are linked to cerebral lesion burdens[45–48]. Notably, these five measures were negatively correlated with the aortic distensibilities (the difference between the maximum and minimum cross-sectional areas divided by the product of the minimum area and the difference between the maximum and minimum blood pressures[24]), serving as a functional proxy for compliance (which measures the vessel's change in volume divided by the change in pressure). Interestingly, the carotid IMT only correlated strongly with LD.

These strong inverse correlations indicate that enlargement of large elastic arteries often coexists with loss of elasticity. These alterations increase transmission of pulsatile energy to fragile small vessels, promoting WMH formation due to small brain vessel injury[47–49]. Crucially, these correlations (but not those for IMT with aortic areas) persisted after adjustment for hypertension, suggesting that they are a general signature of vascular remodeling, driven by endothelial dysfunction, that can also arise from intrinsic factors rather than elevated blood pressure alone[45]. While our correlational analysis does not allow causal inference, our findings are consistent with the prevailing view that atherosclerosis contributes to systemic large-artery stiffening, reflected by reduced

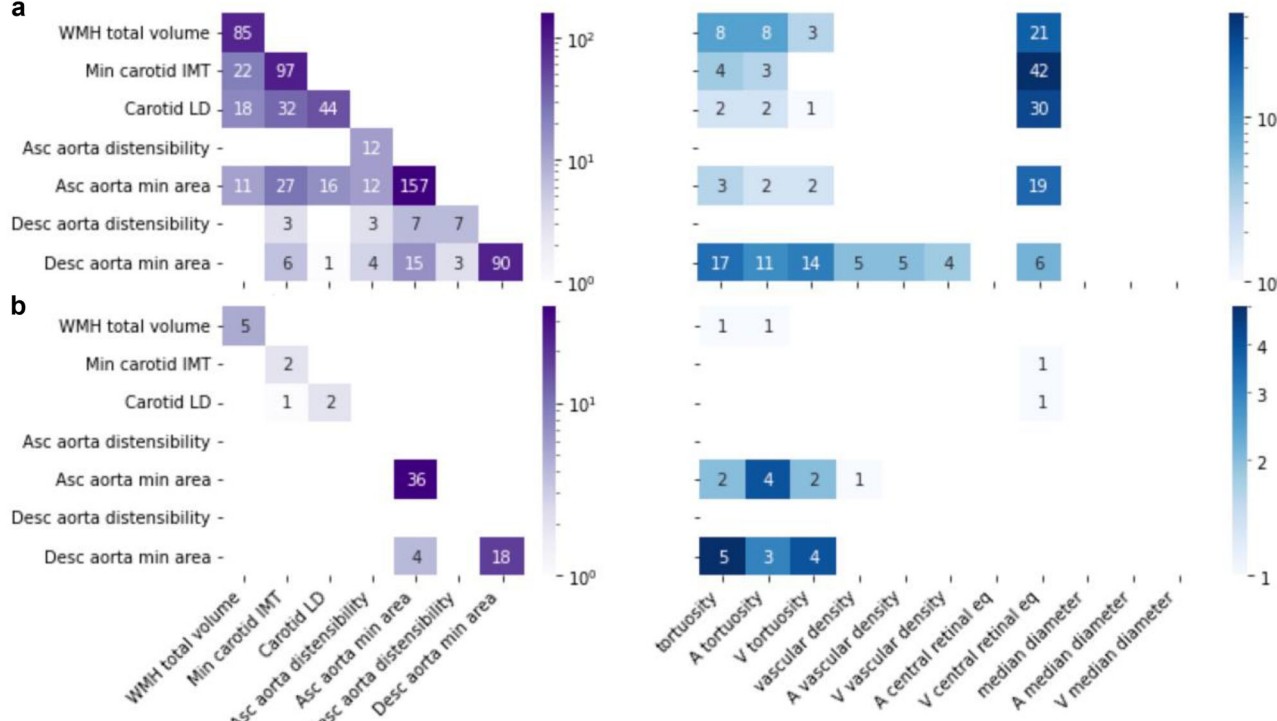

**Fig. 3 | Gene and pathway scoring intersection analysis reveals shared genetic etiology between non-retinal and retinal vascular phenotypes. a** Gene-scoring intersection showing genes in common between non-retinal IDPs and non-retinal IDPs (**left**), and retinal-IDPs (**right**). The diagonal (**left**) shows the number of genes significantly associated with each IDP. While the other cells show the number of intersected genes in IDP pairs. **b** Pathway-scoring intersection showing pathways in common between non-retinal IDPs and non-retinal IDPs (**left**), and retinal-IDPs (**right**). The diagonal (**left**) shows the number of pathways significantly associated with each IDP. While the other cells show the number of pathways in the intersection between pairs of IDPs. Genes and pathways results were obtained using *PascalX*[41].

distensibility, and to compensatory arterial enlargement through positive remodeling[47–49].

Greater WMH volume was associated with narrower diameters and reduced vascular density in retinal arterioles. Since retinal imaging in the UKB was performed on average about 9 years before that of the brain, aorta, and carotid assessments, longitudinal studies are needed to establish causality. Still, our current evidence suggests that retinal microvascular deterioration may precede corresponding cerebral changes, with WMH reflecting ongoing small-vessel disease[50,51]. Importantly, a dedicated sensitivity analysis (Supplementary Note 3) confirmed that these cross-organ associations were robust to the temporal gap between imaging modalities. The principal correlations, including those involving WMH, were almost the same in strength and direction across cohorts with short (≤2 years) and long (up to 10 years) acquisition gaps, suggesting that the temporal gap does not substantially bias the observed associations.

Consistent with this pathomechanistic macro-microvascular link, we also found retinal arteriolar density to be positively correlated with aortic distensibilities and inversely correlated with IMT, LD, and aortic areas. Indeed, larger IMT, LD, and aortic areas, along with reduced aortic distensibilities are indicative of vascular aging and/or pathology[52–55]. Moreover, our finding of positive correlations between arteriolar retinal vessel diameter measurements and aortic distensibilities support previous reports showing that greater aortic stiffness is associated with retinal arteriolar narrowing[56]. Yet, we did not observe a significant correlation between IMT with the retinal diameter size, as suggested in other studies[57]. Our findings indicate that arterial remodeling and functional stiffening act across vascular territories, with downstream effects manifesting as both retinal capillary rarefaction and WMH burden[56,57]. These findings support the concept, advanced by oculomics, that the retina serves as a sensitive marker of systemic microvascular health[58].

It is noteworthy that the phenotypic and genetic correlations of IMT and LD with the other IDPs followed similar patterns, yet those for LD were considerably stronger, despite the much smaller sample size for which we could assess this IDP[23].

The fact that many significant cross-organ phenotypic correlations were either absent or much weaker at the genetic level suggests that they are likely driven by environmental factors such as lifestyle. In contrast, within a single organ, there was typically a strong alignment between phenotypic and genotypic correlations, indicating that the environmental impact is less pronounced in this case.

Nevertheless, the shared associated genes between WMH, IMT, the aorta, and the retinal vascular IDPs suggest common underlying mechanisms, with genes like *EIF3K*, *COL4A2*, and *SMAD3* emerging as promising candidates for modulators of systemic vascular health. These genes appear to converge on complementary pathways involved in vessel wall integrity and stress response. In particular, *SMAD3* coordinates TGF-β-driven vessel-wall maintenance[59], *COL4A2* fortifies the type-IV-collagen basement membrane[60], and *EIF3K* has been implicated in fine-tuning stress-responsive mRNA translation, potentially influencing vascular cell adaptation under conditions of injury or oxidative stress[61]. Furthermore, the identification of shared pathways between retinal and aortic IDPs underscores the interconnected genetic mechanisms that modulate vessel properties across organs. Particularly, the recurrent enrichment for actin pathways pinpoints cytoskeletal remodeling as the common denominator, since the same actin-myosin networks that keep endothelial junctions tight in retinal capillaries also sustain contractile tone and wall strength in the aorta, variants that weaken these networks can therefore manifest simultaneously as retinal vessel tortuosity and aortic dilatation[62,63].

The generalisability of our findings is limited by the specificity of the IDPs studied and the population characteristics of the UKB cohort. While compelling, our results may not fully translate to diverse populations or

those with different health profiles. Practical limitations also include the resolution constraints of non-invasive imaging modalities and the potential for selection bias in the UKB cohort, and the temporal gap between CFP and brain, aorta, and carotid assessments, which limits causal inference. Our study primarily focused on genetic and phenotypic correlations, which can only serve as a starting point to elucidate the mechanistic pathways driving them. Future research should aim to validate our findings in more diverse populations and explore the mechanistic pathways underlying the observed associations. In addition, other local genetic correlation approaches can be used, such as LAVA[64], which could provide complementary information.

Our study provides initial insight into relationships between vessel properties at different scales, from the microvasculature (as reflected in retinal IDPs and WMH as a marker of small vessel health) to the macrovasculature (properties of the aorta and carotid). This understanding can enhance our knowledge of systemic connections between various vasculatures and aid in the development of better prognostic tools for non-retinal vascular diseases using non-invasive, cost-effective retinal imaging.

## Data availability

Phenotypic data from the UKB are available upon application through the UKB website: https://www.ukbiobank.ac.uk. GWAS summary statistics used for this study can be found in: 'Aorta summary statistics', 'Retina summary statistics', and 'Carotid summary statistics'. Figshare https://doi.org/10.6084/m9.figshare.30521345.v1and can be found under the following link https://figshare.com/articles/journal_contribution/Morphological_vascular_IDPs_measured_from_UKBB_images/30521345?file=59268953.

## Code availability

The code for this study is available on the public GitHub repository: https://github.com/BergmannLab/multiorgan_vascular_IDPs.

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

## Acknowledgements

The authors thank Leah Böttger, Olga Trofimova, David Presby, and Dennis Bontempi for their insightful comments on the manuscript. We also extend our gratitude to the study participants and the staff of the UKB, as well as the researchers who measured the IDPs and made the GWAS summary statistics publicly available. This research has been conducted using the UK Biobank Resource under Application Number 90947. Supported by the Swiss National Science Foundation grant no. CRSII5_209510 for the "VascX" Sinergia project. The authors declare no conflicts of interest.

## Author contributions

S.O.V. conceived and designed the study under the supervision of S.B. S.O.V. reviewed available summary statistics from vascular IDPs in the UKB, harmonized the data, and developed the analysis pipelines. S.O.V. performed all statistical analyses, including covariate adjustment, construction of the cross-organ IDP correlation map, heritability estimation, cross-IDP genetic correlation, and PascalX gene- and pathway-level analysis. S.O.V. and S.B. wrote the manuscript and supplementary material.

## Competing interests

The authors declare no competing interests.
