## [Transparent Peer Review file · Communications Medicine]

Cross-organ Analysis Reveals Associations between Vascular Properties of the Retina, the Carotid and Aortic Arteries, and the Brain

Corresponding Author: Professor Sven Bergmann

Version 0:

Reviewer comments:

Reviewer #1

(Remarks to the Author)

Ortín Vela and Bergman describe an examination of phenotypic and genetic correlations among vascular phenotypes from the brain, carotid, retina, and heart. Notably, they find that retinal phenotypes are correlated with imaging phenotypes from the aorta, carotid, and brain, highlighting the retina as a potential biomarker for general vascular health. They also find evidence supporting a shared genetic basis for these traits.

I have several comments, primarily regarding areas where additional detail or justification is needed, as well as a few suggestions for additional results and analyses.

For the phenotypic correlations, all phenotypes were adjusted for several factors, including genotype batch and ancestry principal components (PCs). What was the rationale for these adjustments? Were the phenotypic correlations calculated using individuals of non-European ancestry, or was some restriction based on ancestry applied?

For the retinal IDPs, which eye(s) was selected for analysis? What was the justification for this choice?

The main correlation results are presented in Figure 2:

- The caption states, "Colors indicate standardized effect sizes for linear regressions (a, b) or the genetic correlation coefficient (c)." Is this correct, or do the colors in (a) and (b) represent Pearson's correlation coefficient, as described in the Methods?
- I recommend using the same color scale across these plots for clarity (perhaps a log color scale would be appropriate). For example, within the non-retinal IDP–non-retinal IDP correlations on the left-hand side, the genetic correlations appear larger than they actually are due to the different scale used in plot (c) compared to plots (a) and (b). Similarly, the non-retinal IDP–retinal IDP correlations on the right-hand side appear to be of greater magnitude than they are, due to the very different scale compared to the non-retinal IDP–non-retinal IDP correlations.

What LD scores were used for the LDSC analyses?

A comparison of the observed phenotypic correlations versus genetic correlations for each trait pair could be informative. While this is somewhat described in the text, a plot would allow for easier visualization and comparison.

The authors assess the shared genetic basis of these traits using (1) global genetic correlations and (2) the overlap of genes implicated through PascalX analyses. Significant but modest genetic correlations were found for several trait pairs, while genes common to multiple traits were identified through the PascalX analyses. Could the shared genetic underpinnings be further elucidated using a local genetic correlation approach, such as LAVA (<https://doi.org/10.1038/s41588-022-01017-y>)?

Reviewer #2

(Remarks to the Author)

Comments on Abstract:

Regarding "However, it is not well understood to what extent retinal vascular morphology reflects that in other organs":

This statement appears to be too broad and inaccurate. The paper should more precisely state that it's examining the specific relationships between retinal vascular phenotypes and vascular phenotypes in other organs. Authors could consider revising the wording to be consistent with the description in the Introduction part. That's more accurate.

Regarding the data provided:

The abstract provides insufficient quantitative information. It merely states that relationships exist without specifying their magnitude or significance. More specific data should be included to strengthen the abstract.

Regarding "This highlights the potential of retinal imaging as a non-invasive prognostic tool for systemic vascular health":

This conclusion is not well supported by the preceding correlation data. The abstract should clarify how the observed correlations specifically support the use of retinal imaging as a prognostic tool. Correlation alone doesn't necessarily indicate prognostic value.

Comments on Introduction:

First paragraph:

The description of basic concepts is overly detailed and lengthy. This section should be more concise, focusing only on the essential background information needed to contextualize the research. Consider condensing this significantly.

Terminology:

Replace "color fundus images (CFIs)" with "color fundus photographs (CFPs)" throughout the paper, as this is the more commonly used and accepted terminology in the field.

Comments on Results, Discussion, and Conclusion:

Overall structure and emphasis:

The detailed reporting of results (indicators and relationships) is good but not where the paper's true value lies. The most significant contribution of this paper should be in the discussion section, which needs substantial expansion. The relationships identified and possible mechanisms deserve more thorough examination.

Discussion section recommendations:

Expand the discussion of pathophysiological mechanisms that might explain the observed relationships. Provide more detailed interpretation of what these relationships signify for understanding the connection between retinal findings and systemic conditions. This would have more implications for future clinical applications and research on retina-systemic condition relationships, which is the paper's most important contribution. If the limited discussion is due to word count limitations, authors could consider presenting more results in figures rather than text descriptions.

Reviewer #3

(Remarks to the Author)

This study investigates the systemic associations of vascular morphology across multiple organs, specifically the retina, carotid artery, aorta, and brain, using image-derived phenotypes (IDP) and genetic data from the UK Biobank. The analysis includes phenotypic correlations across organ-specific vascular IDPs, genetic correlations using GWAS summary statistics, and genes and pathways scores with PascalX. The findings suggest that retinal vascular features may serve as indicators of systemic vascular health, positioning retinal imaging as a promising non-invasive proxy for cardiovascular and cerebrovascular conditions. While the paper involves extensive computation and thorough analyses, several concerns need to be addressed.

1) One of the major limitations of this study is the significant temporal gap between the collection of different IDPs, particularly between retinal imaging and other modalities. In the UK Biobank, most retinal images were collected around 2010, whereas carotid ultrasound and MRI data were acquired around 2020. This substantial time difference may considerably affect the validity of the observed correlations at both the phenotypic and genetic levels. The authors should carefully discuss this issue and consider conducting a sensitivity analysis using retinal images collected during later visits

(with a smaller temporal gap) to assess whether similar correlations can still be observed.

2) This study relies exclusively on UK Biobank data. Although the authors acknowledge this limitation in the discussion, the absence of validation using independent cohorts reduces the generalizability and robustness of the findings.

3) While the analyses adjust for hypertension, other critical cardiovascular risk factors that significantly influence vascular IDPs, such as diabetes, smoking, and cholesterol levels, are not accounted for. The omission of these important confounders may bias the correlation results and should be addressed or, at a minimum, discussed.

4) The findings on phenotypic and genetic correlations have not been well discussed. The lack of interpretation provides limited insights. For example, while shared genes such as SMAD3 and COL4A1/2 are identified, their potential biological roles in vascular morphology across different organs are not thoroughly discussed. Similarly, although actin-related pathways are mentioned, their relevance to vascular manifestation or disease mechanisms is not elaborated. A deeper biological interpretation would significantly enhance the impact and insightfulness of the study.

5) Regarding the associations between retinal morphological indices and systemic conditions, an established research area known as Oculomics includes many relevant studies that should be acknowledged and discussed in this work.

Version 1:

Reviewer comments:

Reviewer #1

(Remarks to the Author)

Thanks for the revisions. The extra information is clear and sufficient, and your justifications make sense. I don't have any further comments or concerns.

Reviewer #1

Reviewer #3

(Remarks to the Author)

The authors have addressed all my concerns and substantially improved the manuscript. The added sensitivity analyses, extended adjustment for cardiovascular risk factors, and deeper biological interpretation have strengthened the rigour and robustness of the study. The revisions now provide a clearer view of the systemic mechanisms linking retinal and non-retinal vascular traits. I have no further comments on this paper.

Rebuttal Letter COMMSMED-25-0312:

Reviewers' comments and our responses:

Reviewer #1 (Remarks to the Author):

Ortín Vela and Bergman describe an examination of phenotypic and genetic correlations among vascular phenotypes from the brain, carotid, retina, and heart. Notably, they find that retinal phenotypes are correlated with imaging phenotypes from the aorta, carotid, and brain, highlighting the retina as a potential biomarker for general vascular health. They also find evidence supporting a shared genetic basis for these traits.

I have several comments, primarily regarding areas where additional detail or justification is needed, as well as a few suggestions for additional results and analyses.

For the phenotypic correlations, all phenotypes were adjusted for several factors, including genotype batch and ancestry principal components (PCs). What was the rationale for these adjustments? Were the phenotypic correlations calculated using individuals of non-European ancestry, or was some restriction based on ancestry applied?

The rationale for adjusting our IDPs for these factors is to control for potential confounders that could systematically influence the vascular phenotypes across different organs and potentially lead to spurious inflation of their correlations. By applying the same adjustment strategy to all phenotypes, we aimed to reduce the risk of such inflation.

Regarding ancestry, we did not restrict the analysis to individuals of European ancestry. Our goal was to maximize the sample size while mitigating population structure effects through the inclusion of genetic principal components (PCs). We adjusted for these PCs as a standard approach to account for ancestry-related variation in genetic and phenotypic data.

We have now clarified these aspects in our revised manuscript by adding the following in the *Phenotypic correlation* section (4.3) of our Methods:

The inclusion of genetic PCs in both retinal and non-retinal IDP adjustments aimed to account for population structure without restricting the analysis to individuals of European ancestry. This allowed us to retain a larger and more diverse sample while mitigating the confounding effects of ancestry.

For the retinal IDPs, which eye(s) was selected for analysis? What was the justification for this choice?

Whenever retinal images from both eyes were available, we computed the retinal IDP for both images and took the average; otherwise, we used the IDP from whatever image (right or left) was present in the UK Biobank (UKB).

Our justification for this procedure, which we already used in our previous study, was that taking the average of both eyes tends to reduce measurement noise and improve robustness. We have clarified this in the revised manuscript and added a reference to our previous work in the Methods-Phenotypic correlation section as follows:

Whenever retinal images from both eyes were available, the retinal IDP was computed for each eye, and the average was taken; otherwise, the IDP from the available image (right or left) was used.

The main correlation results are presented in Figure 2:

- The caption states, “Colors indicate standardized effect sizes for linear regressions (a, b) or the genetic correlation coefficient (c).” Is this correct, or do the colors in (a) and (b) represent Pearson’s correlation coefficient, as described in the Methods?

We thank the reviewer for pointing this out. In univariate linear regression, the effect size estimate is $\beta = \text{cov}(x,y)/\text{var}(x) = \text{corr}(x,y) \text{std}(y)/\text{std}(x)$. For normalized response (y) and explanatory (x) variables (i.e., both have unit standard deviation), the effect size is equal to the (Pearson) correlation. We have updated the figure legend for consistency to:

Colors indicate Pearson’s correlation coefficients (a, b) or the genetic correlation coefficient (c).

- I recommend using the same color scale across these plots for clarity (perhaps a log color scale would be appropriate). For example, within the non-retinal IDP–non-retinal

IDP correlations on the left-hand side, the genetic correlations appear larger than they actually are due to the different scale used in plot (c) compared to plots (a) and (b). Similarly, the non-retinal IDP–retinal IDP correlations on the right-hand side appear to be of greater magnitude than they are, due to the very different scale compared to the non-retinal IDP–non-retinal IDP correlations.

We thank the reviewer for this thoughtful feedback. In fact, we also had originally chosen the same color scale for all three subfigures. The reason we decided to use a different color scale for subfigure (c) was that even though the corresponding (absolute) genetic correlations were the largest, none of them was significant (Figure c, right). Using their dynamic range for presenting the phenotypic correlations in subfigures (a) and (b) effectively makes them look less prominent, even though many of them are indeed significant. In particular, applying the same color range across all subpanels, it becomes difficult to distinguish subtle color differences between (a) and (b). For this reason, we would prefer to stick with the current presentation, but we are open to alternative suggestions and have included below a version of our figure using the same color scheme for all three subplots.

What LD scores were used for the LDSC analyses?

We used the 1000G EUR reference panel, and we have added this information to the results section:

Genetic correlations and h^2 were computed using LDSR, using the 1000G EUR reference panel.

A comparison of the observed phenotypic correlations versus genetic correlations for each trait pair could be informative. While this is somewhat described in the text, a plot would allow for easier visualization and comparison.

We thank the reviewer for raising this point. We also thought about having a plot of the phenotypic against the genetic correlations. Yet, we finally did not include this figure since most of the standard errors for the genetic correlation estimates are very large (the values are provided in a Google Sheet linked in the document). As a result, many IDP pairs showed sizable point estimates for genetic correlation but were not statistically significant. This made a visual comparison potentially misleading, as it could overemphasize uncertain estimates. For this reason, we opted not to include such a plot, although we acknowledge its conceptual value.

The authors assess the shared genetic basis of these traits using (1) global genetic correlations and (2) the overlap of genes implicated through PascalX analyses. Significant but modest genetic correlations were found for several trait pairs, while genes common to multiple traits were identified through the PascalX analyses. Could the shared genetic underpinnings be further elucidated using a local genetic correlation approach, such as LAVA (<https://doi.org/10.1038/s41588-022-01017-y>)?

We thank the reviewer for this insightful suggestion regarding the use of local genetic correlation approaches to further investigate shared genetic underpinnings. We fully agree that local genetic correlation analyses can provide complementary information beyond global measures.

After evaluating LAVA and similar methods, we opted to expand our genetic overlap analysis using an approach that aligns more closely with our overall framework and the tools already used in the study, particularly PascalX. Specifically, instead of simply intersecting significant genes identified per IDP (Main figure 3a), we now incorporate genetic correlation directionality to compute coherent and anti-coherent genes between each pair of IDPs. This approach, based on the [PascalX Coherence test], allows us to integrate both gene-level association strength and the sign of genetic correlation.

This new analysis, which is now included in the Supplementary Data Coherent genes, reveals additional shared genes between IDPs, along with the direction of their genetic effects. For instance, WMH and IMT exhibited mostly coherent genes, suggesting aligned genetic influences, whereas IMT and LD showed predominantly anticoherent genes, indicating opposing genetic effects.

Based on these results, we have updated our main text, including the following:

Further analyses using a local genetic correlation approach were conducted to identify shared genes among pairs of IDPs, moving beyond simple intersection of individually associated genes. For this, we employed PascalX cross-GWAS analysis [36], which allowed us to distinguish coherent (positively correlated) and anticoherent (negatively correlated) genetic effects. Notably, WMH and IMT exhibited mostly coherent genes, suggesting aligned genetic influences, whereas IMT and LD showed predominantly anticoherent genes, indicating opposing genetic effects. The venular central retinal equivalent showed exclusively coherent gene relationships with both WMH and IMT, while showing mainly anticoherent associations with LD and the descending aorta minimum area. Associations between tortuosities and the descending aorta minimum area were largely coherent. Detailed data on coherence and anticoherence relationships can be found in Supplementary Data 'Coherent genes'.

We believe this gene-level coherence analysis offers a localized perspective on shared genetic architecture that, while different from the region-based partitioning of LAVA, still provides valuable insights into the functional overlap across organ IDPs. However, we cite this paper now, expanding our limitations paragraph:

In addition, other local genetic correlation approaches can be used, such as LAVA [56], which could provide complementary information.

Final comment to reviewer #1: We sincerely thank the reviewer for the constructive and insightful feedback. Your comments helped us clarify key methodological details, revisit and better explain important choices (such as adjustment strategies and figure design), and prompted us to expand our genetic analysis with a new gene-level coherence approach using PascalX. We appreciate your input, which has meaningfully improved the clarity and depth of the manuscript.

Reviewer #2 (Remarks to the Author):

Comments on Abstract:

Regarding "However, it is not well understood to what extent retinal vascular morphology reflects that in other organs":

This statement appears to be too broad and inaccurate. The paper should more precisely state that it's examining the specific relationships between retinal vascular phenotypes and vascular phenotypes in other organs. Authors could consider revising the wording to be consistent with the description in the Introduction part. That's more accurate.

We agree with this comment and have modified the first part of our abstract to:

Vascular properties of the retina are indicative not only of ocular but also of systemic cardio- and cerebrovascular health. However, the specific relationships between retinal vascular phenotypes and those in other organs have not been systematically investigated in large samples. Here, we compared vascular image-derived phenotypes from the brain, carotid artery, aorta, and retina from the UK Biobank, with sample sizes ranging from 18,808 to 68,740 participants per phenotype.

Regarding the data provided:

The abstract provides insufficient quantitative information. It merely states that relationships exist without specifying their magnitude or significance. More specific data should be included to strengthen the abstract.

We thank the reviewer for this comment. We fully agree and have now explicitly reported the most significant cross-organ correlations observed. While these values were already presented in the figures and tables, we have added them directly to the main text, both in the Abstract and the Results section, for greater clarity:

White matter hyperintensities were positively correlated with carotid intima-media thickness ($r=0.03$), lumen diameter ($r=0.14$), and aortic cross-sectional areas ($r=0.09$), but negatively correlated with aortic distensibilities ($r\leq-0.05$). Arterial retinal vascular density showed negative correlations with white matter hyperintensities ($r=-0.04$), intima-media thickness ($r=-0.04$), lumen diameter ($r=-0.06$), and aortic areas ($r=-0.05$), while positively correlated with aortic distensibilities ($r=0.04$).

Regarding "This highlights the potential of retinal imaging as a non-invasive prognostic tool for systemic vascular health":

This conclusion is not well supported by the preceding correlation data. The abstract should clarify how the observed correlations specifically support the use of retinal imaging as a prognostic tool. Correlation alone doesn't necessarily indicate prognostic value.

We agree that our findings are entirely correlational and that we should avoid overstating prognostic implications. Accordingly, we rephrased the sentence of our abstract to:

Our study sheds light on the complex interplay between vascular morphology in different organs, revealing shared and distinct genetic underpinnings, and suggesting that retinal vascular features may reflect broader vascular morphology.

Comments on Introduction:

First paragraph:

The description of basic concepts is overly detailed and lengthy. This section should be more concise, focusing only on the essential background information needed to contextualize the research. Consider condensing this significantly.

We thank the reviewer for this helpful comment. We were uncertain about the appropriate level of detail, so we appreciate this feedback. In response, we have revised the introduction by shortening the paragraph on basic vascular physiology, merging and trimming the modality descriptions, and condensing the cardiovascular associations. For example, we shortened the first paragraphs from:

The vascular system is a complex network of blood vessels, essential for circulating blood throughout the body. Blood flows from the heart first through arteries, branching out to smaller arterioles, that feed a vast network of capillaries, delivering oxygen and nutrients to various tissues and collecting waste products. Venules and veins return blood to the heart, and act as volume reservoirs maintaining a pressure gradient crucial for blood circulation.

Simple physiological measures such as arterial blood pressure and blood oxygen saturation provide valuable insights into the functionality of the vascular system. However, medical imaging offers more detailed information specific to different vascular structures.

To:

The vascular system ensures the distribution of oxygen and nutrients and the removal of waste through a hierarchical network of arteries, capillaries, and veins. While basic physiological measures, such as blood pressure, provide global insights into vascular function, medical imaging enables the quantification of detailed, organ- and vessel-specific phenotypes.

Terminology:

Replace "color fundus images (CFIs)" with "color fundus photographs (CFPs)" throughout the paper, as this is the more commonly used and accepted terminology in the field.

We were not aware that "color fundus photographs (CFPs)" is the more widely accepted term. We have now updated the manuscript accordingly, replacing all instances of "CFIs" with "CFPs".

Comments on Results, Discussion, and Conclusion:

Overall structure and emphasis:

The detailed reporting of results (indicators and relationships) is good but not where the paper's true value lies. The most significant contribution of this paper should be in the discussion section, which needs substantial expansion. The relationships identified and possible mechanisms deserve more thorough examination.

Discussion section recommendations:

Expand the discussion of pathophysiological mechanisms that might explain the observed relationships.

Provide more detailed interpretation of what these relationships signify for understanding the connection between retinal findings and systemic conditions. This would have more implications for future clinical applications and research on retina-systemic condition relationships, which is the paper's most important contribution. If the limited discussion is due to word count limitations, authors could consider presenting more results in figures rather than text descriptions.

We thank the reviewer for highlighting the need for a deeper interpretation. Accordingly, we have expanded the discussion of pathophysiological mechanisms to explain the associations:

[...] In this study, we found that also morphological measures of different arteries, specifically the cross-sectional areas of both the ascending and descending aorta, as well as the carotid LD, are all positively correlated with each other. We also observed positive correlations between WMH volume and carotid measures, further supporting previous reports that greater IMT and LD are linked to cerebral lesion burdens [37–40]. Notably, these five measures were negatively correlated with the aortic distensibilities (the difference between the maximum and minimum cross-sectional areas divided by the product of the minimum area and the difference between the maximum and minimum blood pressures [24]), serving as a functional proxy for compliance (which measures the vessel's change in volume divided by the change in pressure). Interestingly, the carotid IMT only correlated strongly with LD.

These strong inverse correlations indicate that enlargement of large elastic arteries often coexists with loss of elasticity. These alterations increase transmission of pulsatile energy to fragile small vessels, promoting WMH formation due to small brain vessel injury [39–41]. Crucially, these correlations (but not those for IMT with aortic areas) persisted after adjustment for hypertension, suggesting that they are a general signature of vascular remodeling, driven by endothelial dysfunction, that can also arise from intrinsic factors rather than elevated blood pressure alone [37]. While our correlational analysis does not allow causal inference, our findings are consistent with the prevailing view that atherosclerosis contributes to systemic large-artery stiffening, reflected by reduced distensibility, and to compensatory arterial enlargement through positive remodeling [39–41].

[...]

Consistent with this pathomechanistic macro–microvascular link, we also found retinal arteriolar density to be positively correlated with aortic distensibilities and inversely correlated with IMT, LD, and aortic areas. Indeed, larger IMT, LD, and aortic areas, along with reduced aortic distensibilities are indicative of vascular aging and/or pathology [44–47]. Moreover, our finding of positive correlations between arteriolar retinal vessel diameter measurements and aortic distensibilities support previous reports showing that greater aortic stiffness is associated with retinal arteriolar narrowing [48]. Yet, we did not observe a significant correlation between IMT with the retinal diameter size, as suggested in other studies [49]. Our findings indicate that arterial remodeling and functional stiffening

act across vascular territories, with downstream effects manifesting as both retinal capillary rarefaction and WMH burden [48-49].

[...] the shared associated genes between WMH, IMT, the aorta, and the retinal vascular IDPs suggest common underlying mechanisms, with genes like EIF3K, COL4A2, and SMAD3 emerging as promising candidates for modulators of systemic vascular health. These genes appear to converge on complementary pathways involved in vessel wall integrity and stress adaptation. In particular, SMAD3 coordinates TGF- β -driven vessel-wall maintenance [51], COL4A2 fortifies the type-IV-collagen basement membrane [52], and EIF3K has been implicated in fine-tuning stress-responsive mRNA translation, potentially influencing vascular cell adaptation under conditions of injury or oxidative stress [53]. Furthermore, the identification of shared pathways between retinal and aortic IDPs underscores the interconnected genetic mechanisms that modulate vessel properties across organs. Particularly, the recurrent enrichment for actin pathways pinpoints cytoskeletal remodeling as the common denominator, since the same actin-myosin networks that keep endothelial junctions tight in retinal capillaries also sustain contractile tone and wall strength in the aorta, variants that weaken these networks can therefore manifest simultaneously as retinal vessel tortuosity and aortic dilatation [54, 55].

Final comment to reviewer #2: We thank the reviewer for the detailed and constructive comments, which greatly improved the accuracy and depth of our manuscript. In particular, your feedback helped enhance the discussion by having a deeper exploration of the pathophysiological mechanisms and genetic underpinnings behind the observed associations.

Reviewer #3 (Remarks to the Author):

This study investigates the systemic associations of vascular morphology across multiple organs, specifically the retina, carotid artery, aorta, and brain, using image-derived phenotypes (IDP) and genetic data from the UK Biobank. The analysis includes phenotypic correlations across organ-specific vascular IDPs, genetic correlations using GWAS summary statistics, and genes and pathways scores with PascalX. The findings suggest that retinal vascular features may serve as indicators of systemic vascular health, positioning retinal imaging as a promising non-invasive proxy for cardiovascular and cerebrovascular conditions. While the paper involves extensive computation and thorough analyses, several concerns need to be addressed.

1) One of the major limitations of this study is the significant temporal gap between the collection of different IDPs, particularly between retinal imaging and other modalities. In the UK Biobank, most retinal images were collected around 2010, whereas carotid ultrasound and MRI data were acquired around 2020. This substantial time difference may considerably affect the validity of the observed correlations at both the phenotypic and genetic levels. The authors should carefully discuss this issue and consider conducting a sensitivity analysis using retinal images collected during later visits (with a smaller temporal gap) to assess whether similar correlations can still be observed.

We thank the reviewer for raising this important point. Indeed, the retinal IDPs we used in our analysis were derived in our previously published study based on:

- Instance 0: Initial assessment visit (2006–2010); 67,655 participants
- Instance 1: First repeat assessment visit (2012–2013); 19,420 participants

In contrast, IDPs from other modalities (e.g., carotid ultrasound and MRI) used in the current study primarily correspond to:

- Instance 2: Imaging visit (2014+)

To evaluate the impact of potentially substantial temporal gaps, we conducted an additional sensitivity analysis. Specifically, we re-computed the main Figure 2a Right (reproduced above) using only participants from Instance 1 (shown below), which narrows the gap between retinal imaging and other modalities. Although this reduces the available sample sizes for any given trait pair, most of the highly significant correlations based on instance 0 and 1 (Figure 2a reproduced above), also remained significant using only data from instance 1 with the same sign and similar magnitude, e.g., WMH and arteriolar vascular density ($r = -0.04$ in the main figure, $r = -0.06$ using only instance 1); Carotid lumen diameter and arteriolar vascular density ($r = -0.06$ in the main figure, $r = -0.06$ using only instance 1); Ascending-aorta distensibility and central retinal equivalent ($r = 0.06$ in the main

figure, $r=0.07$ using only instance 1). The heat-map of all pairwise correlations is now provided as a new Supplementary section: Sensitivity to Imaging–Acquisition Date (and below).

As for the genetic correlations, although germline genetic variation is of course stable over time, the phenotypic expression of genetic burden could nevertheless vary with age. Thus, in principle, since later imaging instances effectively phenotype an older cohort, the corresponding IDPs may be more sensitive to genetic modulation that is age-related. However, given that the phenotypic correlations were largely consistent between the smaller and larger temporal gaps, we do not expect a substantial impact of this effect on our GWAS results.

As suggested by the reviewer, we now explicitly refer to this issue in our discussion and the limitations paragraph:

Since retinal imaging in the UKB was performed on average about 9 years before that of the brain, aorta, and carotid assessments, longitudinal studies are needed to establish causality. Still, our current evidence suggests that retinal microvascular deterioration may precede corresponding cerebral changes, with WMH reflecting ongoing small-vessel disease [42, 43]. Importantly, a dedicated sensitivity analysis (Supplementary section ‘Sensitivity to Imaging–Acquisition Date’) confirmed that these cross-organ associations were robust to the temporal gap between imaging modalities. The principal correlations, including those involving WMH, were almost the same in strength and direction across cohorts with short (≤ 2 years) and long (up to 10 years) acquisition gaps, suggesting that the temporal gap does not substantially bias the observed associations.

[...] Practical limitations also include [...] and the temporal gap between CFP and brain, aorta, and carotid assessments, which limits causal inference.

2) This study relies exclusively on UK Biobank data. Although the authors acknowledge this limitation in the discussion, the absence of validation using independent cohorts reduces the generalizability and robustness of the findings.

We agree that external replication is important for establishing generalizability. At present, however, we are not aware of an available independent cohort that (i) includes the full combination of imaging modalities needed to derive our vascular IDPs (carotid ultrasound, cardiac/aortic MRI, brain MRI for WMH, and retinal imaging), (ii) has comparable image-processing pipelines or raw data access enabling harmonized IDP generation, and (iii) attains sufficient sample size after applying the quality-control steps required for these phenotypes. Consequently, a direct replication was not feasible within the scope of this work.

We have taken several steps to facilitate future replication. All analytic code and phenotype definitions required to reproduce the UK Biobank analyses will be made available (see Data and Code Availability). We also provide detailed variable mappings (UKB field IDs) and preprocessing descriptions so that investigators with access to suitably rich multimodal datasets can implement comparable pipelines.

3) While the analyses adjust for hypertension, other critical cardiovascular risk factors that significantly influence vascular IDPs, such as diabetes, smoking, and cholesterol levels, are not accounted for. The omission of these important confounders may bias the correlation results and should be addressed or, at a minimum, discussed.

We appreciate this important point. Our primary analyses focused on vascular imaging phenotypes rather than disease states; we therefore initially used hypertension as a broad indicator of vascular burden to avoid over-stratifying and to retain statistical power. Now, in response to this comment, we conducted additional sensitivity analyses in which we incrementally adjusted for a more comprehensive cardiovascular risk set. Specifically, we fit models with the following covariate blocks: basic covariates (age, age², sex, 10 genetic PC), hypertension status, diabetes, smoking, and HDL cholesterol.

We regenerated the correlation heatmaps (new Supplementary Figure 4 and shown below) using the extended risk set and compared them with the original Main Fig. 2b. The principal findings are unchanged: all of the strongest associations remain directionally consistent and of very similar magnitude. A small number of weaker associations moved above or below the multiple-comparison significance threshold, but effect sizes changed only modestly. No new large correlations emerged after fuller adjustment.

We have added these analyses and a paragraph in the Results noting that the results are robust to additional adjustment for diabetes, smoking, and HDL cholesterol:

To evaluate whether additional cardiovascular risk factors might further confound these relationships, we repeated the analyses including an extended covariate set with diabetes, current smoking, and HDL cholesterol, together with the previous covariates and hypertension status. Results remained consistent with the hypertension-adjusted analyses: effect sizes changed only modestly, and all of the previously observed strong correlations persisted (Supplementary Figure 4).

Main figure 2b: covariates + hypertension:

Supplementary figure 4: covariates + hypertension + diabetes +smoking + HDL:

4) The findings on phenotypic and genetic correlations have not been well discussed. The lack of interpretation provides limited insights. For example, while shared genes such as SMAD3 and COL4A1/2 are identified, their potential biological roles in vascular morphology across different organs are not thoroughly discussed. Similarly, although actin-related pathways are mentioned, their relevance to vascular manifestation or disease mechanisms is not elaborated. A deeper biological interpretation would significantly enhance the impact and insightfulness of the study.

We thank the reviewer for highlighting the need for a deeper interpretation. We have expanded the discussion to better interpret the phenotypic and genetic correlations by elaborating on the potential biological roles of key genes and pathways involved in vascular morphology:

[...] In this study, we found that also morphological measures of different arteries, specifically the cross-sectional areas of both the ascending and descending aorta, as well as the carotid LD, are all positively correlated with each other. We also observed positive correlations between WMH volume and carotid measures, further supporting previous reports that greater IMT and LD are linked to cerebral lesion burdens [37–40]. Notably, these five measures were negatively correlated with the aortic distensibilities (the difference between the maximum and minimum cross-sectional areas divided by the product of the minimum area and the difference between the maximum and minimum blood pressures [24]), serving as a functional proxy for compliance (which measures the vessel's change in volume divided by the change in pressure). Interestingly, the carotid IMT only correlated strongly with LD.

These strong inverse correlations indicate that enlargement of large elastic arteries often coexists with loss of elasticity. These alterations increase transmission of pulsatile energy to fragile small vessels, promoting WMH formation due to small brain vessel injury [39–41]. Crucially, these correlations (but not those for IMT with aortic areas) persisted after adjustment for hypertension, suggesting that they are a general signature of vascular remodeling, driven by endothelial dysfunction, that can also arise from intrinsic factors rather than elevated blood pressure alone [37]. While our correlational analysis does not allow causal inference, our findings are consistent with the prevailing view that atherosclerosis contributes to systemic large-artery stiffening, reflected by reduced distensibility, and to compensatory arterial enlargement through positive remodeling [39–41].

[...]

Consistent with this pathomechanistic macro–microvascular link, we also found retinal arteriolar density to be positively correlated with aortic distensibilities and inversely correlated with IMT, LD, and aortic areas. Indeed, larger IMT, LD, and aortic areas, along with reduced aortic distensibilities are indicative of vascular aging and/or pathology [44–47]. Moreover, our finding of positive correlations between arteriolar retinal vessel diameter measurements and aortic distensibilities support previous

reports showing that greater aortic stiffness is associated with retinal arteriolar narrowing [48]. Yet, we did not observe a significant correlation between IMT with the retinal diameter size, as suggested in other studies [49]. Our findings indicate that arterial remodeling and functional stiffening act across vascular territories, with downstream effects manifesting as both retinal capillary rarefaction and WMH burden [48-49].

[...] the shared associated genes between WMH, IMT, the aorta, and the retinal vascular IDPs suggest common underlying mechanisms, with genes like EIF3K, COL4A2, and SMAD3 emerging as promising candidates for modulators of systemic vascular health. These genes appear to converge on complementary pathways involved in vessel wall integrity and stress adaptation. In particular, SMAD3 coordinates TGF- β -driven vessel-wall maintenance [51], COL4A2 fortifies the type-IV-collagen basement membrane [52], and EIF3K has been implicated in fine-tuning stress-responsive mRNA translation, potentially influencing vascular cell adaptation under conditions of injury or oxidative stress [53]. Furthermore, the identification of shared pathways between retinal and aortic IDPs underscores the interconnected genetic mechanisms that modulate vessel properties across organs. Particularly, the recurrent enrichment for actin pathways pinpoints cytoskeletal remodeling as the common denominator, since the same actin-myosin networks that keep endothelial junctions tight in retinal capillaries also sustain contractile tone and wall strength in the aorta, variants that weaken these networks can therefore manifest simultaneously as retinal vessel tortuosity and aortic dilatation [54, 55].

5) Regarding the associations between retinal morphological indices and systemic conditions, an established research area known as Oculomics includes many relevant studies that should be acknowledged and discussed in this work.

We thank the reviewer for suggesting this inclusion. We included this now explicitly, both in the introduction and discussion, respectively:

[...] changes in the retinal vasculature have been linked to vascular issues in distant organs, including stroke [9–11], coronary heart disease [12, 13], or hypertension [14, 15], highlighting systemic links central to oculomics, an emerging field that uses retinal imaging to assess overall systemic health [16].

[...] These findings support the concept, advanced by oculomics, that the retina serves as a sensitive marker of systemic microvascular health [50].

Final comment to reviewer #3: We are grateful for the reviewer's targeted feedback, which helped us to identify and address potential gaps in both analysis and interpretation. This input also encouraged us to clarify the underlying systemic vascular mechanisms more comprehensively. In particular, conducting the new sensitivity analysis on the temporal gap between imaging modalities was invaluable in reinforcing the robustness and validity of our findings.

Rebuttal Letter COMMSMED-25-0312:

Reviewers' comments and our responses:

Reviewers' comments:

Reviewer #1 (Remarks to the Author):

Thanks for the revisions. The extra information is clear and sufficient, and your justifications make sense. I don't have any further comments or concerns.

Reviewer #1

Reviewer #3 (Remarks to the Author):

The authors have addressed all my concerns and substantially improved the manuscript. The added sensitivity analyses, extended adjustment for cardiovascular risk factors, and deeper biological interpretation have strengthened the rigour and robustness of the study. The revisions now provide a clearer view of the systemic mechanisms linking retinal and non-retinal vascular traits. I have no further comments on this paper.

We sincerely thank you for your valuable comments and suggestions during the previous review stages. We are delighted to hear that you feel we have addressed all your concerns and that the revisions have substantially improved the rigor and robustness of the manuscript, particularly concerning the sensitivity analyses, extended adjustments for cardiovascular risk factors, and biological interpretation.

We are very pleased that the changes implemented have fully met your requests and now provide a clearer view of the systemic mechanisms linking retinal and non-retinal vascular traits.